# A Simulation Model for the Inductor of Electromagnetic Levitation Melting and Its Validation

**DOI:** 10.3390/ma16134634

**Published:** 2023-06-27

**Authors:** Błażej Nycz, Roman Przyłucki, Łukasz Maliński, Sławomir Golak

**Affiliations:** Department of Industrial Informatics, Silesian University of Technology, ul. Akademicka 2A, 44-100 Gliwice, Poland

**Keywords:** electromagnetic levitation melting, metal melting

## Abstract

This article presents a numerical model of electromagnetic levitation melting and its experimental validation. Levitation melting uses the phenomenon of magnetic induction to float a melted, usually metallic, conductor in an electromagnetic field. With the appropriate configuration of the coil (the source of the alternating magnetic field), the eddy currents induced in the molten batch interact with the coil magnetic field, which causes the melted metal to float without direct contact with any element of the heating system. Such a contactless process is very beneficial for melting very reactive metals (e.g., titanium) or metals with a high melting point (e.g., tungsten). The main disadvantage of levitation melting is the low efficiency of the process. The goal of the authors is to develop, by means of a numerical simulation and optimization tools, a system for levitation melting with acceptable efficiency. To achieve this, it is necessary to develop a reliable and representative computational model. The proposed model includes an analysis of the electromagnetic field, with innovative modeling of the convective heat transport. Experimental validation of the model was performed using aluminum alloy, due to the lack of the need to use a protective atmosphere and the ease of measurements. The measurements included electrical values, the melted batch positions during levitation, the melting time, and the temperature distribution in its area. The verification showed that the compliance between the computational model and the simulation for the position of the batch was accurate to 2 mm (6.25%), and the consistency of the batch melting time was accurate to 5 s (5.49%). The studies confirmed the good representativeness of the developed numerical model, which makes it a useful tool for the future optimization of the levitation melting system.

## 1. Introduction

### 1.1. Electromagnetic Levitation Melting

Technological progress requires the use of more and more perfect materials. However, their use is often hindered by the price and technological difficulties associated with the processing of new materials. Typical examples of this type of material are refractory metals. Depending on the adopted definition, these include niobium, molybdenum, tantalum, tungsten, and rhenium, as well as titanium, vanadium, chromium, manganese, zirconium, ruthenium, rhodium, hafnium, osmium, and iridium [1,2]. These are characterized by a high melting point, high mechanical strength, and general wear resistance. They are widely used in pure form [3,4] but also as alloys [5,6] or alloy additions [7]. These materials are becoming more and more widely used, resulting in the search for cheaper methods of producing these metals and their alloys, as well as technologies for manufacturing the final products. The basic advantage of these metals (i.e., refractority) is also a serious problem in their processing. Currently, all methods of melting these metals are based on electricity. The following techniques are used: arc remelting (remelted material electrode) [8,9,10,11], plasma melting [10,12], electron beam melting [10,13,14], melting in a cold crucible [15,16], inductive melting [17], and melting using electromagnetic levitation [18,19]. Some metals, such as titanium, are particularly difficult to heat treat because they are very reactive at high temperatures, leading to contamination. Currently, metals and alloys with high strength and high melting temperatures are increasingly used in many applications.

Due to its high strength and low density, titanium and its alloys are used in aviation and space technologies (as a construction material for airframes and capsules) [20], as well as in machine construction and military technologies. In addition, the high temperature resistance of titanium makes it suitable for use in the construction of aircraft engines (turbine blades) [21], and its biocompatibility makes it suitable for use in healthcare [22,23]. In cases where extreme temperature resistance is required (nuclear fusion reactors, rocket and jet engines, and generally plasma-facing material), tungsten alloys are used despite their high density [24,25]. The use of titanium and its alloys has been a long-known technological problem, mainly related to machining and heat treatment [26]. The source of these problems is not only its high hardness and negligible thermal conductivity but also its high melting point and high reactivity at high temperatures [27,28]. The latter property makes it difficult to maintain titanium purity during heat treatment, and it is difficult to obtain titanium alloys with a precise composition [29]. Titanium alloys are obtained on a large scale in arc furnaces and on a slightly smaller scale in induction furnaces with a cold crucible [30,31], but in both cases, only part of the batch is high purity, whereas the rest is contaminated [32]. A cold hearth is used to reduce titanium contamination in arc remelting, and a similar technique is used for cold crucible induction melting [33]. The melted titanium is in contact with the intensively cooled base of the furnace, where its surface layer solidifies and forms a titanium coating that prevents further contamination. This solution, in addition to material losses, also causes technological difficulties if melting is a preliminary stage before the casting process. Namely, due to intensive cooling, it is difficult to achieve the required degree of overheating of the batch.

By its very nature, a process that allows melting without contact with the “environment” is melting using electromagnetic levitation [34,35,36]. Magnetic levitation is based on the interaction of two alternating magnetic fields (similarly, as two opposite magnetic poles repel each other). The sources of the alternating magnetic field are the exciter current (the coils supplying the entire system) and the eddy currents induced in the batch. The exciter current creates an alternating magnetic field, which induces eddy currents in the batch. These currents, in turn, create a magnetic field that is opposite to that of the exciter, with the result that they repel each other. The metal (conductor) floats inside the inductor without contact with other elements. The induction of eddy currents inside the metal creates a magnetic field, but of course, ohmic losses also occur, which causes the material to heat up. Sometimes, to speed up the melting process, additional energy sources are used, such as plasma or electron reheating [37].

The main issue with this type of melting is its generally low energy efficiency; therefore, improvement and optimization of this process are needed. Electromagnetic levitation melting takes place in an induction coil of a specific design (called an inductor) powered by a high-frequency source [38,39,40]. The efficiency of the process is determined mainly by the shape of the inductor and the electrical parameters of the inductor supply, which must be adjusted to the mass, shape, and material properties of the melted batch [41,42]. Currently, numerical modeling [39,43,44] and optimization can be used to find the optimal solution [45,46,47,48]. After developing a numerical model, it should be validated by measurements, which is described herein.

There are many benchmark problems available in the literature, such as the TEAM 35 and TEAM 36 benchmark problems. This benchmark has a good theoretical and measurement basis to compare results or consider different methodologies [49,50]. Comparison of the model with the results obtained by other researchers may be an initial stage of verification, but the reliability of such a comparison depends on the similarity of the models. Benchmark 35 used in [49] is an optimization standard, but it is for a constant magnetic field and modeled as 2D (axisymmetric). In our case, the magnetic field is alternating. The patterns used in [50] also apply to polyoptimization, and in a field quite distant from our field (superconductivity benchmark 22) and die press with electromagnet benchmark 25. The closest model for our experiment would be benchmark 28 for electromagnetic levitation, although it is for a completely different geometry. For this reason, benchmark problems were not included in the research presented.

### 1.2. Aim of the Paper

The research presented in the paper relates to the preparation and validation of the simulation model. The inductor model is designed for electromagnetic levitation melting and is suitable for future optimization. The validation of the model is based on the physical inductor and the measurement station, which allows the following features to be obtained:Temperature as a function of time,Batch position during levitation,Current frequency,Voltage,Current.

The main novelties of the presented model are the relationships between the fluid and the air surrounding the molten batch. Air movements are related to the convection caused by the heating batch. Moreover, the model is asymmetric, so it represents the inductor available to us relatively accurately. Another representation of the actual process that has been considered is the change in the electrical parameters of the material with changes in temperature.

In view of future optimizations, the simulation model must provide a compromise between reasonable consumption of computer memory/time and acceptable accuracy. This paper does not yet cover the optimization problem but is an entry point to it.

## 2. Materials and Methods

### 2.1. Measurement Station

The measurement station (Figure 1) was prepared as a combination of enough tools to measure the different required features. It contained the following:Thermal camera (infrared)-PI 640 (marked 1 in Figure 1). The manufacturer was Optris, and the production location was Germany.Video camera-D5300 (marked 2 in Figure 1). The manufacturer was Nikon, and the production location was Taiwan.Glass tube (marked 3 in Figure 1). The manufacturer and production location are unknown.Current probe with converter CWT 60xB (marked 4 in Figure 1). The manufacturer was Powertek, and the production location was the UK.Oscilloscope-THS720P Handheld Digital Oscilloscope. The manufacturer was Tektronix, and the production location was the USA.

The dimensions of the thermal camera were 45×56×90 mm, and its weight was below 400 g; so, it could easily be mounted above the inductor. Importantly, its temperature measurement range (150 to 900 ${}^{\circ }$C) was sufficient to record the melting of aluminum. The emissivity in the camera was set to 0.02 to match the emissivity of the aluminum batch [51]. The thermal camera recorded the batch from the top. From this position, there were no obstacles, and the camera was a safe distance from the heat source.

The frame rate of the video camera ranged between 24 and 60 fps, and the effective pixels, 24.2 million pixels, were sufficient to record the position of the batch and its shape deformation during melting. The camera was in the front of the inductor mounted on the tripod. The frame included the inductor and ruler, which allowed the digital picture to be scaled.

The tube was made of high-temperature-resistant transparent glass. It was placed in the middle of the inductor to ensure the separation of the wire from the batch. The tube was open from the top; so, the field of view of the thermal camera was not obscured. Moreover, the pictures taken from the video camera facing the front were undisturbed because of the transparency of the glass.

The oscilloscope voltage probe was connected to the ends of the inductor, and it measured the voltage directly on the inductor. The current probe, which was a Rogowski coil with an appropriate transducer, was fastened on the inductor, and the current probe measured the current flowing through the inductor. The current inducted in the coil was calculated as in (Equation 1). The Rogowski coil sketch is shown in Figure 2. The wire inside the loop was the inductor’s wire.
(1)$$u\left(t\right)=M\cdot d_{i}/d_{t},$$
where:*M*—mutual inductance between the current-carrying conductor and the Rogowski coil,$d_{i}/d_{t}$—the rate of change of the current in the conductor (derivative of the current over time).

### 2.2. Geometry and Current Parameters of the Batch and the Inductor

The choice of inductor on which measurements were taken was limited by the options available to the researchers. The current conditions were chosen based on research on a similar inductor [43] from which levitation was possible under our conditions. The main power source available to the researchers had a maximum active power of 15 kW and a maximum frequency of 300 kHz. The levitation of the batch for the selected inductor and current parameters was experimentally confirmed. We performed the measurements for the batch and the inductor, which are shown in Figure 3 on the left side. To simplify the description, we introduced the cross-sectional sketch of the inductor Figure 4. The batch was made of aluminum, which had a solid sphere with radius $\left(r_{B}\right)$ equal to 3 mm. Aluminum was chosen as the batch material because of its low density and low melting point. Other researchers have also used it [41,42,44,46]. The batch was fed into the inductor from the top using a polyester thread, which burned in place where it was connected to the batch at the beginning of the melting process; then, it was removed.

The inductor wire was made of copper and had the shape of a pipe. The inductor wire turned four times counterclockwise and one time in the opposite direction. The inductor was actively cooled by the water that flowed inside the wire tube.

The geometric parameters of the inductor are given in Table 1.

**Figure 4 materials-16-04634-f004:**
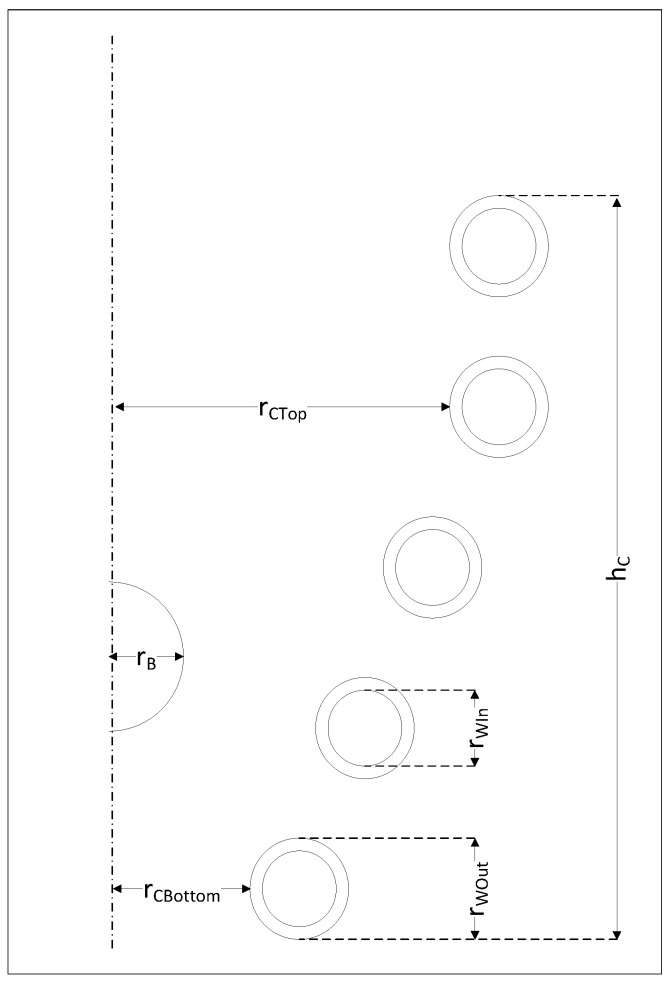
The sketch of a cross section of the inductor. The abbreviations are explained in Table 1.

The electrical parameters measured in the inductor had values, which are summarized in Table 2.

We conducted the above measurements with and without a batch, but the oscilloscope plots did not show significant differences.

### 2.3. Software and Hardware

We modeled the batch and the inductor performing the simulation using Ansys software 20.2. In this study, the following Ansys modules were used:SpaceClaim [52] is a CAD modeling tool that was used to prepare the geometry of the model. It supports a predefined geometry based on mathematical equations, such as a helix, which speeds up the modeling process. Moreover, there is also the possibility of recording models as Python scripts, which allowed us to make more precise and complex modifications to the geometry.Maxwell 3D [53] is a tool that allowed us to simulate the work of electric machines. The Maxwell interface is compatible with the ACIS modeling standard. It supports a common CAD format, including SpaceClaim. Maxwell supports an adaptive mesh generator based on a user-defined convergence level. In addition, the mesh can also be defined by the user. The Maxwell 3D allowed us to calculate the eddy-current effects, assign the current and voltage excitation and boundary conditions (such as the natural boundaries, zero-tangential H Field, Neumann boundaries, and others), conduct the parametric simulation, and many other items.Icepack [53] allows the simulation of heating and cooling of electronic and energetic devices. The temperature of the object can be imported from the simulation result of the other Ansys products or can be assigned directly. The basic solution type is steady state with transient calculations. It also supports temperature changes through convection, radiation, and conduction.Fluent [54] enabled us to calculate the flow of the fluid that had to be simulated in the convection. Fluent has a specific version for user groups with specific needs or constraints. The Fluent simulation can be coupled with other products.

The computer used to simulate the model had an Intel i7-3770 3.4 GHz CPU, 16 GB RAM, and 1800 GB of disk space. We used the 64-bit Windows 10 operating system. The computational capability of the hardware available restricted us from using too complex a model, especially because it will be used for optimization purposes, which certainly requires a large number of subsequent simulations. Therefore, the computation time could not be too long, or the usefulness of the model would be very limited.

### 2.4. Simulation Model

#### 2.4.1. Flow of Simulation

By combining our knowledge of the inductor parameters with the available software, we prepared a simulation model. The first step was to map the inductor geometry itself. The geometry of the model is shown in Figure 5. The model had the following features:

The batch was a solid sphere with a radius of 3 mm.The Z-axis was set at the center of the coil.The inductor in the model had two ends of the wire in the Y-direction, which corresponded to the wire connected to the power source in the physical inductor.In the Y-direction, the wire deformation was visible, which allowed us to change the coil direction of the last wire.In the modeled inductor, the distances between circumvolutions were maintained as the same, but in the physical inductor, they were slightly irregular.

The software used was coupled in the following way. We modeled the geometry in SpaceClaim, and with the model, we prepared the Python script to generate it. The electromagnetic properties we setup in Maxwell 3D. Finally, we applied the heating setup to Icepack with the Fluent Solver. In summary, the data flow between these products is shown in Figure 6. Maxwell 3D obtained the modeled geometry from SpaceClaim and calculated the lifting force for the position of the batch on the Z-axis between 10 to 40 mm. The electromagnetic calculations were carried out for highest position, where the gravity force was equal to the ascending force. Maxwell 3D worked with Icepack in two-way coupling. The electromagnetic loss in the batch was sent to Icepack, which was converted to temperature, and the new temperature of the batch was sent to the Maxwell 3D, which changed the electrical conductivity of the batch.

#### 2.4.2. Electromagnetic Field Analysis

For the introduced model, the following electromagnetic setup was added. For the solution type of the model, we activated the eddy effect for the batch and the inductor (Equation 2), (Equation 3), (Equation 4), and (Equation 5). We also set the calculation type as symbolic. We used a description of the electromagnetic field using a magnetic vector potential *A* and a scalar potential *V* using the symbolic descriptions (Equation 2), (Equation 3), (Equation 4), and (Equation 5). We set the initial temperature to 22 ${}^{\circ }$C, according to the room temperature (Figure 1) during the measurements.

For the aluminum batch, the electrical conductivity was 37,126,300 S/m and changed with the temperature according to the hardcoded equation of the material. The electrical conductivity of the copper inductor was set to 58,000,000 S/m, and due to the active cooling of the water, the thermal modifier was disabled. The convective heat transfer coefficient (α) and the emissivity of the body (ϵ) for each type of material were consistent with the data from the Materials Library from Ansys release 20.2.

The environment during the calculations was air, which was modeled in the shape of a box (Figure 7). The dimensions of the environment were 150×120×80 mm. The faces of the environment had boundary rules set as natural boundaries, which implies that the magnetic field was parallel to the boundary (Equation 6). The boundary conditions for the surface E at the place where the ends of the inductor reached them were different. Here, a current of opposite direction was assigned for both contact surfaces. The assigned current was 340 A; moreover, it alternated with the 277,777 Hz frequency.

We enabled an adaptive mesh for the model, but for the batch, we set the skins with depth according to the penetration depth for this material and frequency as in (Equation 7). Moreover, we set a denser mesh for the inductor to increase the number of elements in the cross section of the wire.
(2)$${▽}^{2}\;A-j\omega \sigma \mu A=J_{s},$$
where
(3)$$J_{s}=\sigma \mu ▽V_{e},$$
(4)$$I_{s}=\mu \underset{S}{\;\int \int \;}J_{s}dS,$$
(5)$$▽\cdot ▽V_{e}=0,$$
(6)n·▽A=0,
(7)$$\delta =\sqrt{2/(\omega *\pi *\sigma *\mu )},$$
where

ω—angular frequency,δ—field penetration depth,σ—electric conductivity,μ—magnetic permeability.$J_{s}$—source current density,$I_{s}$—source current,S—surface,A—magnetic vector potential,j—imaginary unit,ω—angular frequency,σ—conductivity,μ—magnetic permeability,$V_{e}$—electric potential.

The electromagnetic field analysis was based on (Equation 2), (Equation 3), (Equation 4), and (Equation 5).

#### 2.4.3. Thermal Field Analysis

Based on the results of the electromagnetic calculations, thermal analysis was performed (Equation 8). For this purpose, the following features were set as a heating setup. Because of the active cooling of the inductor, we could omit its analysis for the temperature changes. This allowed us to adjust the complexity of the model and to analyze the temperature changes only in the batch and the air surrounding it. The solution type that we set to transient was due to the importance of the heating process in time. We set the initial temperature to 22 ${}^{\circ }$C, which was equal to the temperature for the electromagnetic calculations.

The surroundings of the batch (Figure 8) had a box shape of 13×13×26 mm. The space at the top of the batch was two times larger than the space below the batch. The type of material in the environment was air. We set the boundary conditions on the faces of the surroundings as natural boundaries, which implied that the temperature outside the box was constant.

The electromagnetic losses were imported from the results of the electromagnetic calculations and converted to induction power (Equation 9) and (Equation 10). We enabled temperature changes by convection and radiation (Equation 11) and (Equation 12). Radiation was emitted into a half space. The discrete ordinal radiation model (DO) solved the radiation transfer equation (RTE) for a finite number of discrete solid angles, each of which was associated with a fixed vector direction. The DO model solved as many transportation equations as directions [55]. Radiative heat transfer was assigned to the surface area of the batch. To properly adjust the convection flow, we set gravity in the Z-direction. We enabled the two-way coupling with the electromagnetic part.
(8)$$k{▽}^{2}T+q=\rho c\frac{dT}{dt},$$
(9)J=−jωσμA,
(10)$$q=\frac{1}{2}\frac{{\left|J\right|}^{2}}{\sigma },$$
(11)$$-k\frac{dT}{dn}=\alpha \left(T-T_{a}\right),$$
(12)$$-k\frac{dT}{dn}=\epsilon \sigma \left(T^{4}-T_{a}^{4}\right),$$
where

*k*—thermal conductivity,*T*—temperature,*q*—volume density of the heat source,ρ—mass density,*c*—specific heat capacity.

#### 2.4.4. Fluid Dynamics

The heating of the batch through eddy current induction also affected the system. The heat of the batch (Equation 8) also penetrated the surrounding air, resulting in the convective movement of the air. By reproducing this in the model, it became an even better representation of reality. The heat transfer was modeled using fluid dynamics calculated based on the following equations:(13)$$\frac{d\rho }{dt}+▽\cdot \left(\rho \nu \right)=0,$$
(14)$$\frac{d(\rho \nu )}{dt}+▽\cdot \left(\rho \nu \nu \right)=-▽p+▽\cdot \bar{\bar{\tau }}+\rho g,$$
(15)$$\bar{\bar{\tau }}=\eta \cdot \left(▽\nu +▽\nu^{T}\right)-\eta \frac{2}{3}▽\nu I,$$
(16)$$\frac{d(\rho W)}{dt}+▽\cdot \left(\nu \left(\rho W+p\right)\right)=▽\cdot \left(\lambda_{ef}▽T+{\bar{\bar{\tau }}}_{ef}\cdot \nu \right),$$
(17)$$W=h-\frac{p}{\rho }+\frac{\nu^{2}}{2},$$
where

ρ—fluid density,ν—velocity vector,*p*—static pressure,*g*—gravitational acceleration,η—molecular viscosity,*I*—unit tensor,ρW—energy density,$\lambda_{ef}$—effective conductivity consisting of the conductivity and the conductivity due to the turbulence.

Equation (Equation 13) is the mass conservation equation, also called the continuity equation; the second equation is the momentum conservation Equation (Equation 14). Equation (Equation 16) is the equation of energy conservation in the system. Flow field analysis was carried out using numerical methods. Flow modeling encounters many difficulties. One of them is the modeling of systems with turbulent flows. Therefore, in order to carry out such an analysis directly, based on Equations (Equation 13) to (Equation 16), for example, using the direct numerical simulation method, very dense meshes should be used (number of nodes proportional to $Re^{\frac{9}{4}}$).

In order to reduce the number of variables and close the system of equations, an appropriate turbulence model should be introduced. The two-equation model [54] introduces two additional equations that are often used: kinetic energy transport k and turbulence kinetic energy dissipation rate transport ϵ. These equations require the introduction of additional empirical coefficients, but they allowed us to close the system of equations that describe the motion of the fluid. From the *k*-ϵ model [41], the effective viscosity can be determined by substituting Equation (Equation 19) into Equation (Equation 18):(18)$$\eta_{ef}=\eta +\eta_{t},$$
(19)$$\eta_{t}=\rho C\frac{k^{2}}{\epsilon },$$
where

$\eta_{ef}$—effective viscosity,η—dynamic viscosity,$\eta_{t}$—turbulent viscosity,*k*—kinetic energy of turbulence,ϵ—turbulence energy dissipation coefficient,*C*—empirical constant.

The main advantage of the k-ϵ model is that it gives very good results for many realistic flows of technical importance.

As for the thermal field analysis, the surroundings of the batch (Figure 8) had a box shape of 13×13×26 mm. The space at the top of the batch was two times larger than the space below the batch. The type of material in the environment was air. We set the boundary conditions on the faces of the surroundings (Figure 8) as open, letting the air flow through them.

## 3. Results

### 3.1. The Ascending Force

During the measurements, the batch levitated inside the inductor in a stable position during and after melting. The ruler visible in Figure 4 was at the same distance from the video camera as the melted batch. It was used to estimate the number of pixels in the image that corresponded to 1 mm in reality. The height of the batch levitation was measured from a digital image with a resolution of 1920 × 1080. One millimeter corresponded to nine pixels in the image. From this, the distance of the batch from the bottom scroll was determined. The standard deviation was 1 mm. The results of the measurements are summarized in Figure 9.

The vertical position of the batch during the measurements was 32 mm. For the simulated model, we searched for the position in which the batch levitated. The ascending force was calculated according to (Equation 20) and (Equation 21). For this purpose, we calculated the ascending force for the batch at positions ranging from 10 mm from the bottom of the inductor to 40 mm. The values of the ascending force are shown in Figure 10. The gravity force acting on the batch (marked as a red dashed line) was around 0.003 N; so, there were two positions, namely, at 17 and 30 mm, where these forces compensated one another (equilibrium points), thus allowing the batch to levitate.
(20)fEM=12ReJ×B∗,
(21)$$F_{\mathrm{EM}}=\int_{V}f_{\mathrm{EM}}\;\mathrm{dV}.$$

At position 17 mm, lowering the batch resulted in a decrease in the ascending force, and the batch fell out of the inductor. However, increasing the batch position increased the ascending force, and the batch started to rise. As a consequence, this point represented the unstable equilibrium point. At the 30-mm position, the lower position of the batch increased the ascending force, and the lower position of the batch decreased the ascending force. This implies that this position was a stable equilibrium point and, for this position, further calculations should be carried out [56]. The calculated batch position 30 was 2 mm lower than the batch position registered by the camera. This difference could be derived from the inductor deformations, which were not modeled. A comparison of the levitation position of the batch observed during measurements and that calculated during the simulation is shown in Figure 11.

### 3.2. Batch Heating

We calculated the heating time of the batch in the inductor based on the records from the video and thermal cameras. We merged both recordings into one frame, as shown in Figure 3. The video camera specified the time at which the batch was fed into the inductor, and the thermal camera recorded the moment of melting of the batch. Based on these records, we present the results in Figure 12. The median and average of the melting time measurements were both 91 s. The standard deviation was 3.7497 s.

As a result of the simulations, the melting temperature was obtained in 86 s. The distribution of heat on the batch surface is shown in Figure 13. The temperature was higher in places, where the batch was closer to the inductor wire. The convection effect is visible in Figure 14. The heated air rose and the cold air from the bottom took its place. The simulation heating reached the median melting time 5 s faster in comparison to the measurements. The source of the problem could be an imperfect modeling of the inductor, impurities in the material from which the inductor and batch were made, or fluctuations in the current parameters. As shown in Figure 12, the repeatability of the measurement itself had a standard deviation of 3.46 s. Differences between simulation and measurements were probably caused by the high dynamics of the levitation melting phenomenon. During the levitation, the batch oscillated around the equilibrium point, and these stochastic oscillations were likely to cause different melting points.

### 3.3. Mesh Quality

The accuracy of the simulation calculations depended on the assumptions made to simplify the model, but for the numerical simulations, the accuracy also depended on the correctness of the model’s discretization.

Maxwell provides the possibility of different mesh strategies; the initial mesh prepared by the user is usually homogeneous and may consist of too few elements. To prevent this, an adaptive mesh has been introduced, which performs an analysis before starting the actual simulation. The user can specify the minimum number of mesh densifications that must be made, while the program estimates the error that the adopted mesh introduces into the result. In the case of the model under consideration, we set that at least two mesh compactions took place before the right mesh was determined. In the end, three densifications were performed.

Regardless of the program’s built-in mechanisms, we conducted an independence analysis. Since the purpose of the model was to simulate the temperature changes in the batch while it was in the levitation state, the independence analysis was carried out for two values: the Lorentz force acting on the batch (Figure 15) and the EM losses in the batch (Figure 16). The mesh density for which most of the calculations in this article were made is marked in red on the graphs (Figure 15 and Figure 16). From this reference mesh density, two more models with reduced mesh density and three models with increased mesh density were prepared. The mesh density was increased and decreased in the charge and inductor area, as the most sensitive areas to such changes, because of the depth of penetration of the electromagnetic field. Using a denser grid did not lead to significantly different results, but it did result in a higher use of computer resources, including increased simulation time. The relative deviation between the results in the publication and those obtained for the densest grid was 0.79% for the power and 0.58% for the Lorentz force, with the calculation time increasing from 50 min for the variant in this article to 1 h 20 min for the variant with the densest grid. According to this, we state that the mesh quality was sufficient for the problem considered. The total number of mesh elements was 670,603, and the mesh type was tetrahedral.

### 3.4. Comparison with Other Models

The need to prepare computational models is widespread among researchers, but there is no universal model that is both useful to specific situations and addresses all researchers’ needs. The model presented here complements previous approaches by taking into account the changes in material properties when heating the material, the fluid dynamics in the air surrounding the batch, and the asymmetry of the inductor model.

The previous computational model that we prepared [43] was modeled using gmesh and solved with the getdp solver. With this model, it was possible to calculate the batch losses, the process efficiency, and the forces acting on them. However, it did not take into account the changes in material properties due to temperature changes. Another advantage of that model was its asymmetric mapping of the available inductor. Unfortunately, it was not possible to validate this due to its incompatibility with the cooling system.

There are other similar models, but they differ in some aspects.

In their work, Witteveen et al. [57] proposed a 3D asymmetric model to simulate inductor operation, including the effect of asymmetry on the magnetic field and ascending force. This was verified by comparison with a model published by another author. However, the ability to track the changes in batch temperature over time while taking into account changes in material properties was lacking compared to our model. In addition, the environment of the batch was not taken into account (i.e., it was always assumed that the process took place in a vacuum).

Royer et al. [47] prepared a 2D model, but the lack of asymmetry made the model a simplistic representation of reality. The advantage of this approach was that the position of the batch during the process could be estimated analytically. Furthermore, the model did not take into account changes in material properties for different temperatures and the airflow around it. Heat loss occurs through radiation and convection.

Furthermore, Kermanpur et al. [45] introduced a 2D model, which was also symmetric. The calculations were performed both cyclically and in two stages. First, a harmonic analysis was performed, followed by a thermal analysis. In successive iterations, changes in the properties of the material resulting from the changes in temperature were taken into account. The paper did not provide information on the airflow in the model; so, it was most likely not present. The model was validated experimentally.

Sassonker and Kuperman [44] proposed an electromechanical model that consisted of an electrical part consisting of a series resonant circuit and a typical second-order mechanical subsystem. The model itself was 2D and cylindrical. However, it lacked feedback between the temperature and material properties, and the flow of gas surrounding the batch was not considered. Experimental validation was presented in the paper.

Other applications of numerical modeling related to electromagnetic levitation melting can be found in the literature, which are difficult to compare with the presented model because of their different purposes. The first subtype of such models involves convective flow inside the melted batch [58,59,60]. The second subtype deals with the shape of the molten batch and its oscillations [61,62,63]. The third, on the other hand, is about temperature fields and phase transformations [64,65,66].

The summary of selected properties of simulation models found in the literature is in the Table 3.

## 4. Conclusions and Future Research

Parameters of the simulation model were introduced from the measurement station where the inductor was examined. Based on the geometry of the available inductor, we prepared a geometric model and a strongly coupled electromagnetic–temperature computational model. The following conclusions were drawn:The simulated melting time takes 86 s, which is 5 s faster than the median measured time.Batch levitation occurs at the 30 mm position, which is 2 mm lower compared to the measurements.We investigated the quality of the model by increasing the number of mesh elements and comparing the results. We state the solution quality as good.The computational time of the model is approximately 50 min, which is acceptable for future predictive uses.

The purpose of the proposed simulation model is to perform calculations to examine the impact of the change in the geometry of the melting process. Moreover, the geometry of the inductor should be optimized to increase the efficiency of the melting.

## Figures and Tables

**Figure 1 materials-16-04634-f001:**
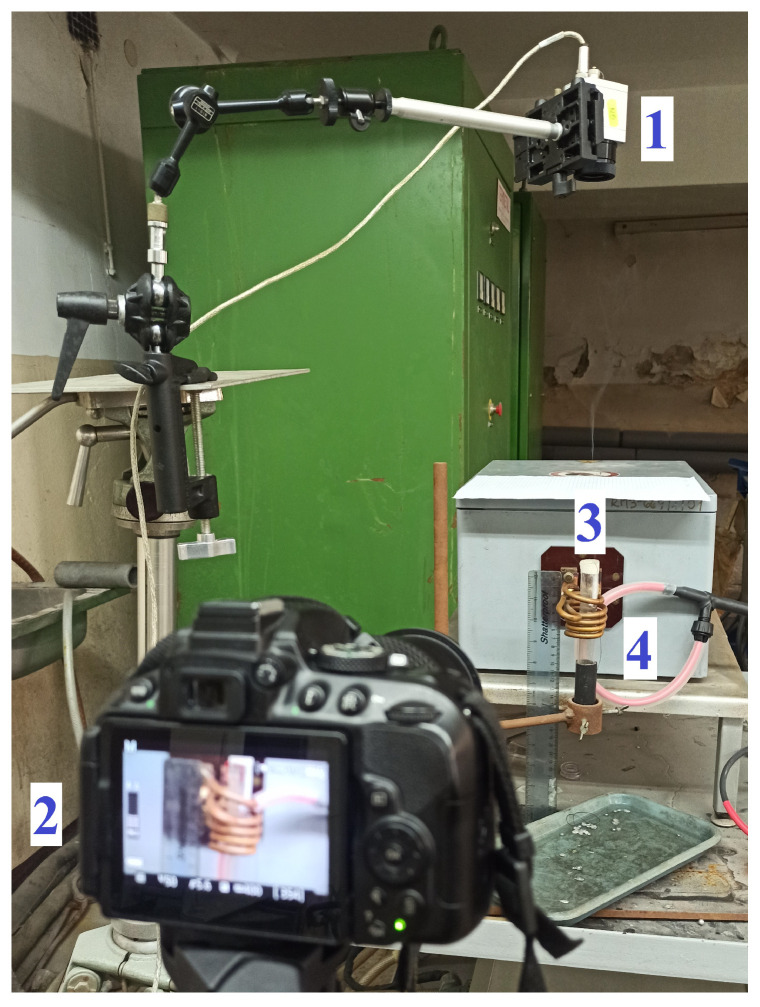
The station used to measure the position and temperature of the batch.

**Figure 2 materials-16-04634-f002:**
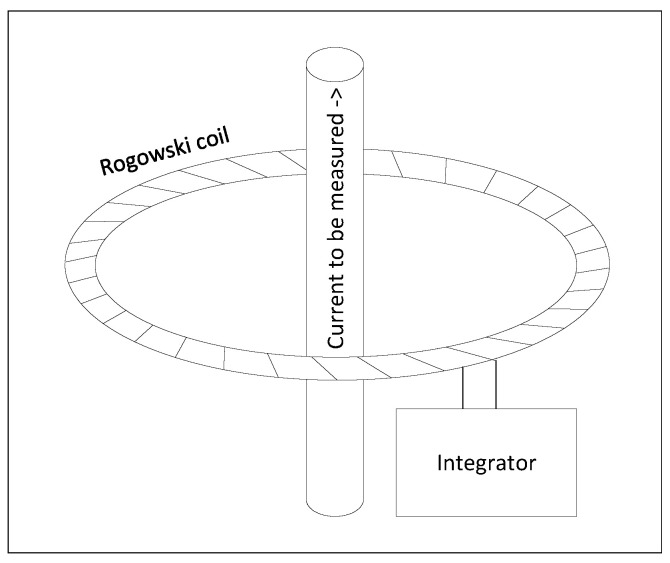
Sketch of the Rogowski coil used to measure the current in the inductor. The Rogowski coil is shown in Figure 1 from 4.

**Figure 3 materials-16-04634-f003:**
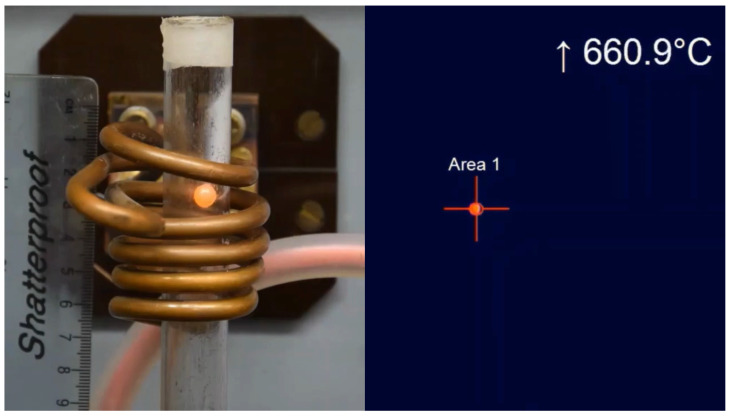
Frame number 12,282 from the video camera (**left**) and the thermal video (**right**). One frame took 0.008 s; so, the images were 98.256 s after the batch was entered.

**Figure 5 materials-16-04634-f005:**
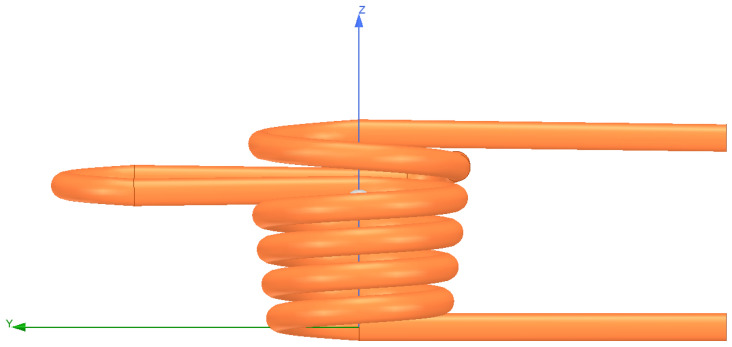
Inductor geometry model prepared in the SpaceClaim program for the inductor on which the measurements were made.

**Figure 6 materials-16-04634-f006:**
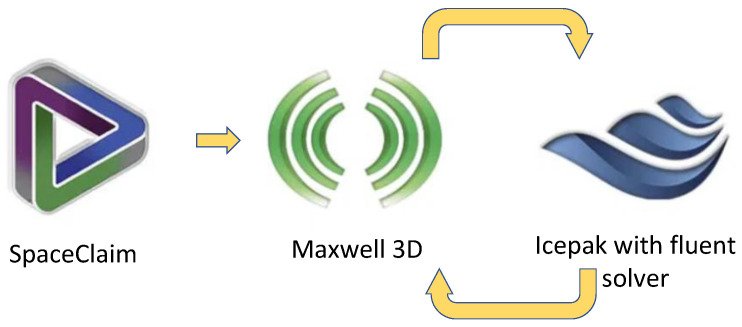
The flow of data between software.

**Figure 7 materials-16-04634-f007:**
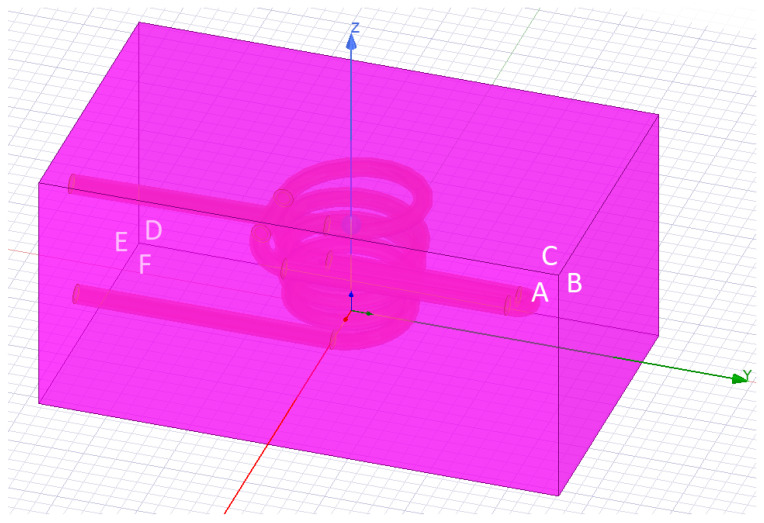
The environment of the electromagnetic model in which the calculations were performed. The surfaces of the environment are marked in amaranth. The letters label the surfaces.

**Figure 8 materials-16-04634-f008:**
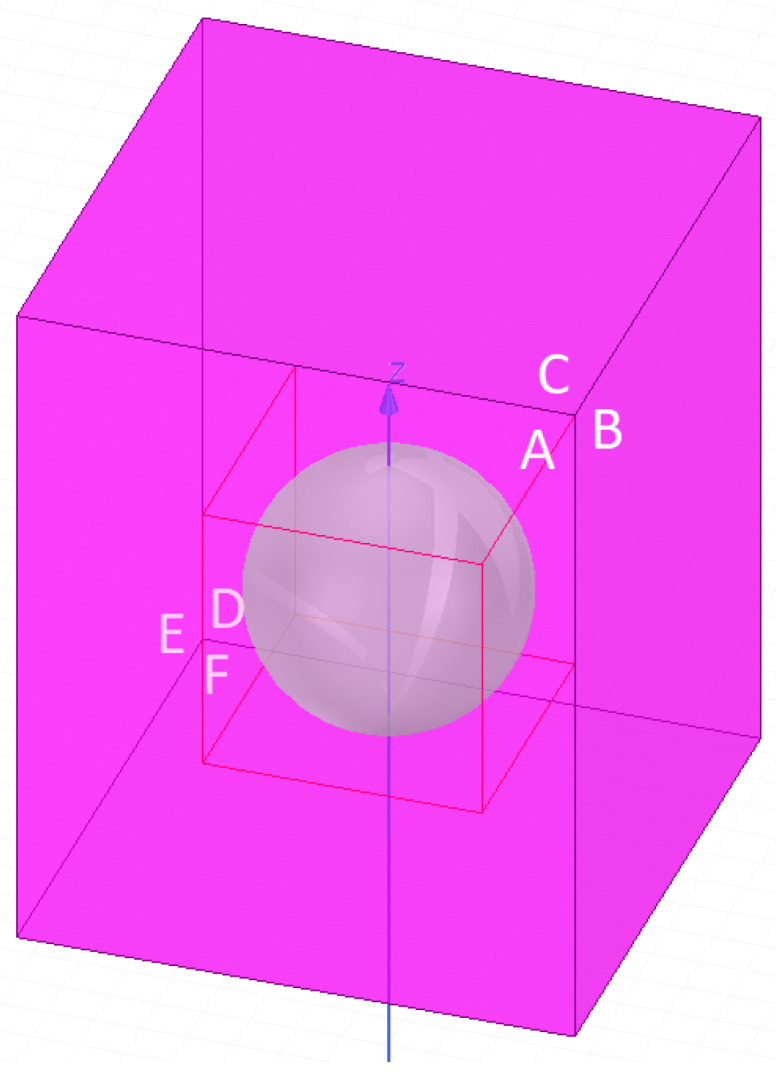
The environment of the thermal–fluid model in which the calculations were performed. The surfaces of the environment are marked in amaranth. The letters label the surfaces.

**Figure 9 materials-16-04634-f009:**
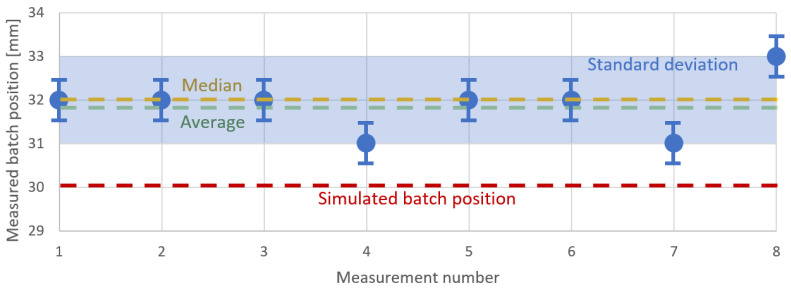
Vertical position of the batch during the process of individual measurements. The yellow dotted line shows the median batch position, and the green dotted line shows the average batch position. The red dotted line shows the batch position calculated from the calculation model. The light blue box shows the standard deviation.

**Figure 10 materials-16-04634-f010:**
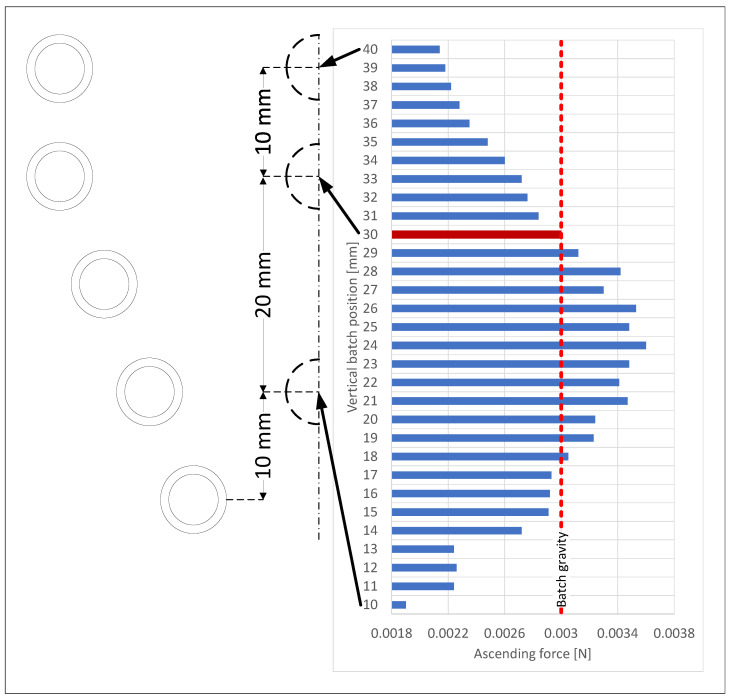
The ascending force acted on the batch (horizontal axis) depending on the height at which the batch was located (vertical axis). The red dashed line is the gravity force acting on the batch. The dark red bar in the chart is a stable equilibrium point.

**Figure 11 materials-16-04634-f011:**
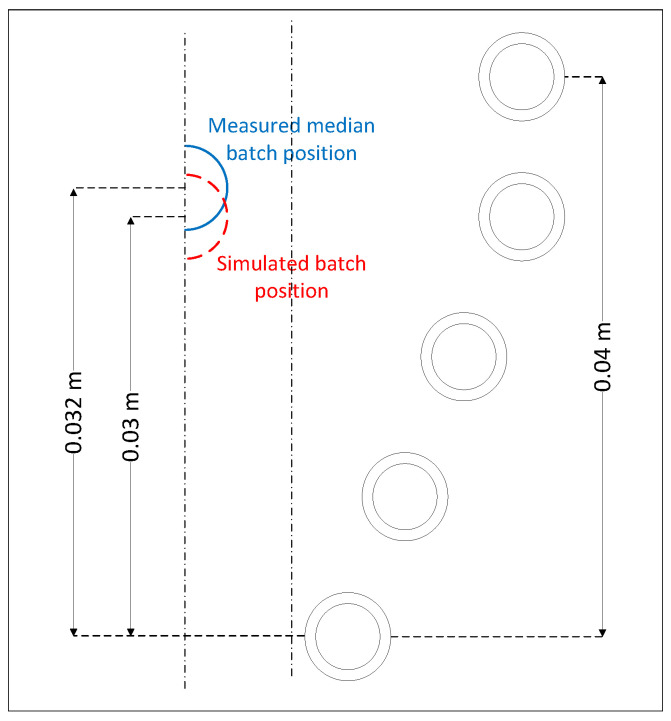
Comparison of the batch levitation positions observed during the measurements and calculated during the simulation. The positions from the measurements are colored blue, and the positions from the simulation are colored red.

**Figure 12 materials-16-04634-f012:**
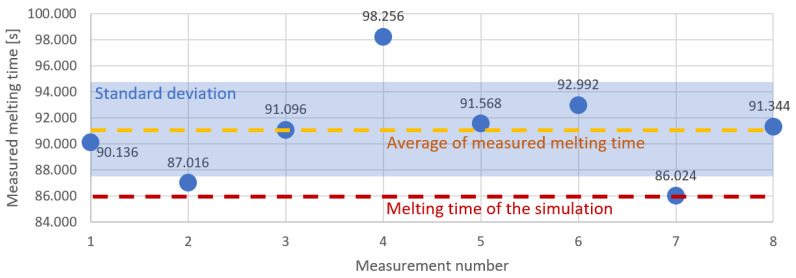
Comparison of successive measurements and the simulation result for the melting times of the batch. The measurements are shown in blue, and the simulation result is shown as a red dashed line. The light blue box shows the standard deviation. The average measured times are represented by the yellow dashed line. Time was measured based on the number of frames between the introduction of the batch and the time the melting temperature was reached. The camera frame rate was 125 Hz; so, the accuracy of the measurement was 0.008 s, making it invisible in the figure.

**Figure 13 materials-16-04634-f013:**
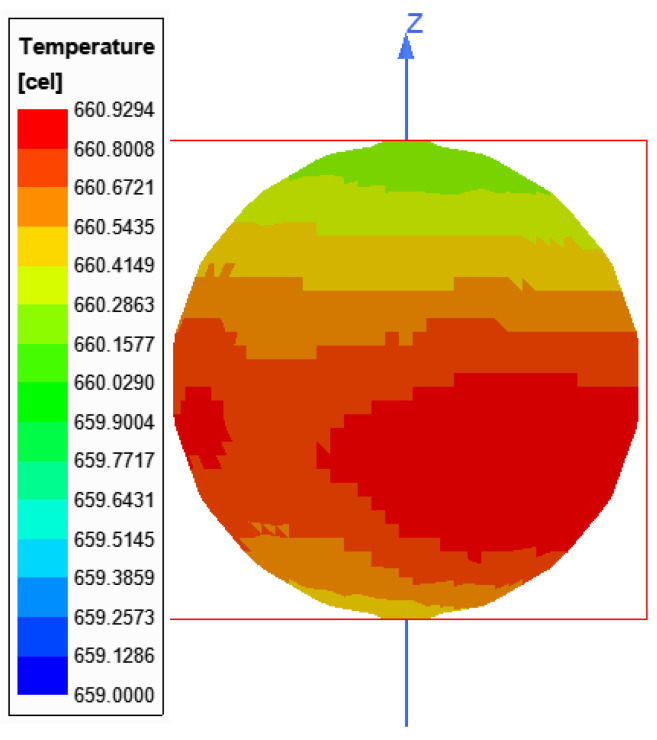
Batch heating simulation 86 s from the start of the process.

**Figure 14 materials-16-04634-f014:**
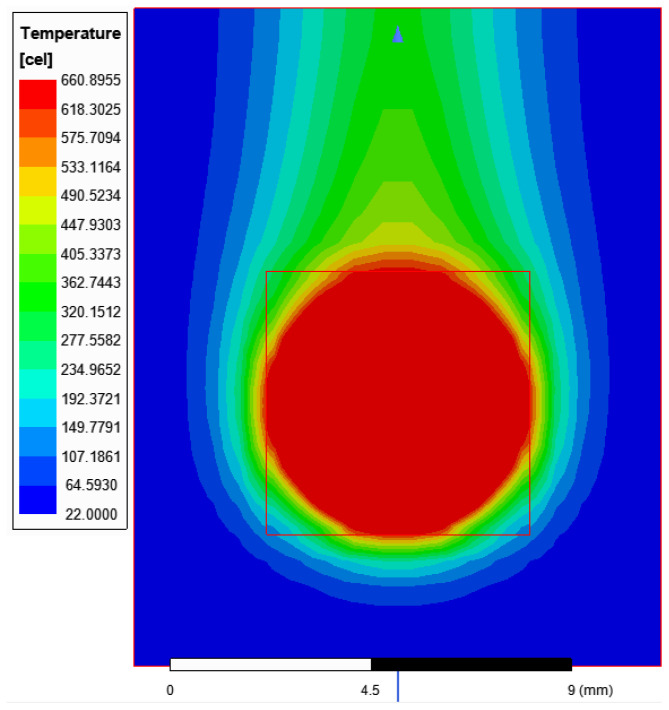
Simulation of heating the batch and the surrounding air 86 s from the start of the process. The convection effects are visible.

**Figure 15 materials-16-04634-f015:**
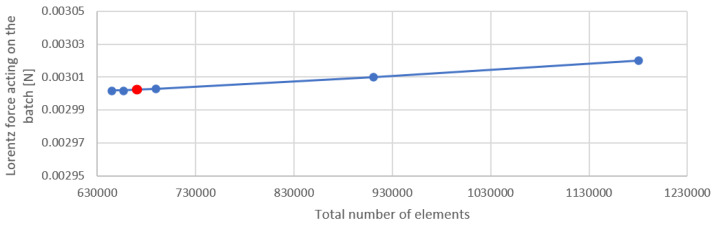
Analysis of the sensitivity of the Lorentz force for the batch with the change in the density of the discretization mesh. The number of mesh elements we used is indicated in red.

**Figure 16 materials-16-04634-f016:**
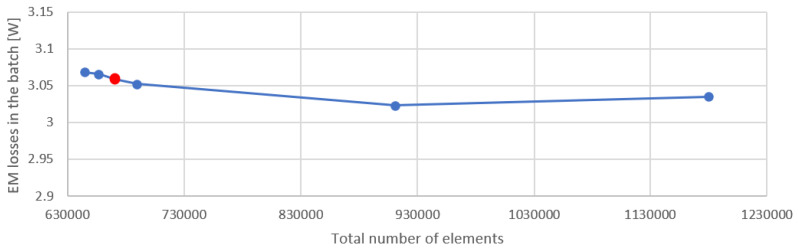
Analysis of the sensitivity of the EM loss for the batch with the change in the density of the discretization mesh. The number of elements we used is indicated in red.

**Table 1 materials-16-04634-t001:** The inductor’s geometric parameters. A graphical representation of each symbol is shown in Figure 4.

Symbol	Meaning	Value [mm]
$r_{WIn}$	The inner radius of the wire	2
$r_{WOut}$	The outer radius of the wire	3
$r_{CBottom}$	The bottom radius of the inductor	14
$r_{CTop}$	The top radius of the inductor	18
$h_{C}$	The total height of the inductor	40

**Table 2 materials-16-04634-t002:** The supply parameters.

Supply Parameter	Value
Current	340 A
Voltage	520 V
Phase shift of current and voltage	$95^{\circ }$
Current frequency	277,777 Hz

**Table 3 materials-16-04634-t003:** The summary of selected properties of simulation models found in the literature.

Model Source	Model Is Asymmetrical	Verification Method	Interaction with Surroundings	Material Property Actualized with Temperature Change
Nycz et al. (2021) [43]	Yes	Not verified	No interaction	No
Witteveen et al. (2021) [57]	Yes	By other model	No interaction	No
Royer et al. (2013) [47]	No	Not verified	Convection and radiation	No
Kermanpur et al. (2011) [45]	No	By experiment	No interaction	Yes
Sassonker and Kuperman (2020) [44]	No	By experiment	No interaction	No
Current paper	Yes	By experiment	Convection and radiation	Yes

## Data Availability

Not applicable.

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
