# Peer review of "A Simulation Model for the Inductor of Electromagnetic Levitation Melting and Its Validation"

_materials, 2023, doi:10.3390/ma16134634_

Round 1

Reviewer 1 Report

The article presents a validation of the simulation model of the inductor to electromagnetic levitation melting. The proposed simulation model is to perform calculations to examine the impact of the change in the geometry of the melting process, however, the geometry of the inductor is not optimized in the current form. Moreover, the proposed simulation model does not closely relate to the fields of Materials. Thus I'd like to suggest the submission of this work to more specialized journals.

Author Response

We would like to thank you for this positive review. We have put a lot of effort into the preparation of our study and we are pleased that it is appreciated. Regarding the selection of the journal, the article will be in the “Manufacturing Processes and Systems” section, which in our opinion is suitable for the metal melting method of Electromagnetic Levitation Melting.

At the same time, we would like to inform you that the new version of this article has been submitted for professional language proofreading and has been certified.

Reviewer 2 Report

A simulation model of the inductor to electromagnetic levitation melting was proposed in this paper, and the results were experimentally verified and analyzed in detail. Overall, this article has a certain degree of innovation, and the workload of this article is also relatively sufficient. However, there are still some formatting issues and unclear description problems, which is shown as follows:

1. Section 2, detailed information about the experimental equipment, such as the manufacturer and production location, needs to be provided.

2. The use of the symbol such as [1] in Figure 1 is not appropriate, which may lead readers to mistake it for referring to a reference.

3. Tables 1 and 2 need to be changed to a three-line table format.

4. Although Table 1 provides the geometric dimensions of the model, it is still necessary to indicate in Figure 5 and Figure 7 which geometric structure each parameter represents.

5. Section 2.4, what grid structure is used in the article? Please provide a figure for the grid structure. What is the number of grids? Where is grid independence analysis?

6. Section 3.4, could you use a table to summarize the differences between the proposed model and other models in terms of calculation time, accuracy, and applicability, etc.?

7. Section 4, the innovation and purpose of this article do not need to be repeated in the conclusion section, only the main conclusion of the article and future research and application directions need to be written.

Minor editing of English language required.

Author Response

We thank you very much for the time you devoted to our article. We can see from your comments that you have thoroughly read our work and we appreciate it. Regarding your comments:

  1. In Section 2, information on the equipment of the measuring station is supplemented with suggested data.
  2. The references to the various parts of Figure 1 have been changed so that they are not confused with citations.
  3. The format of the tables has been changed to a three-line format, we believe that this treatment increases the readability of the data presented.
  4. Table 1 refers to Figure 4. As there was no relevant description there, it could have been confusing for the reader. The reference has been added, in addition, a visualization of some dimensions has been added to Figure 5.

At the same time, we would like to inform you that the new version of this article has been submitted for professional language proofreading and has been certified.

Reviewer 3 Report

The manuscript presents a simulation of levitation melting by inductors. It is clearly written and sincere as to the possible sources of discrepancies between model predictions and measurements. The results will have an impact on future developments in this area.

I would consider recommending the manuscript for publication in Materials, if the following points are addressed in a revised version.

Comments on research
-----------------------

1. Two of the benchmark observables in the study are the batch vertical position and melting point. The discrepancy observed in the former between measurement and simulation is assigned by the authors to the imperfections in the modelled geometry. One way to surpass it could be to modify the positions of the wires, e.g. the three higher ones, in a way to make the calculated value match that of the measured one.

This suggestion does not only aim to reproduce the actual batch vertical position, but to further correct the model as far as the melting time is concerned. It would hopefully lead to a more realistic modelling and a less prominent deviation from experimental values. Of course, it would not affect other possible sources of mis-modelling.

2. In fig. 9, is the melting time measured with such precision that the error bars are not visible?

3. In lines 316-317, individual measurements are discussed that do not add much as to explaining the deviation, so I propose to remove the sentence "but for the second ...."

4. In figures 9 and 12, the reader would benefit from the standard deviation of the measured valued. I suggest to show it as, e.g., a vertical bar and also quantify the simulated value deviation in terms of it.

5. Fig. 13 and fig. 14 captions: they look similar. In fig. 14, the authors could specify that the convection effects are visible.

6. I suggest to use millimetres in distance measurement, e.g. z axis, throughout the text to match the units in the graphs.

Editorial comments:
--------------------

L40: "temperature in the function of the time" do you mean "temperature as a function of time" ?

L51: "Due to future optimisation" -> "In view of future optimisations" ?

L51: "compromise of" -> "compromise between"

L52: "computer memory and computing time" -> "computer memory/time"

L68: emisivity -> emissivity

L71: "The frame of the video camera rate" -> "The frame rate of the video camera ranges"

L88: "M" should be in italics

fig2 caption: rogowski -> Rogowski

L149: do you mean "The Z-axis is set at the centre of the coil" ?

L159: Icepak -> Icepack

L161-163: It is not clear the meaning. Can you rephrase? Also, if 10 refers to Fig. 10, it has to be positioned earlier, as Fig. 7.

L169: "set the eddy effect and" can be omitted

L249: "The first equation used (13)" -> "Equation (13)"

L258: k in italics

General comment: all units should be in upright roman (not italic), e.g. in L65, L67, ...

Author Response

We thank you very much for the time you devoted to our article. We can see from your comments that you have thoroughly read our work and we appreciate it. Regarding your comments:

  1. Thank you for your comments regarding the methodology.
    Our modeling mode was typical, i.e. we performed calculations, and an inductor was made for the model from the calculations. Due to the fact that the geometry of the electromagnetic levitation exciter is not trivial, and the manufacture of the exciter was carried out as a craft, there were deviations from the shape of the project. The solution may be to perform calculations for the real (made inductor), but it requires scanning the inductor with a 3D scanner, which we, unfortunately, do not have. There remains, of course, the problem of other sources of imperfections such as material purity or possible fluctuations in current parameters. Other potential sources of model deviations from the actual inductor were added to the article.
  2. A description has been added to Figure 9 to explain the lack of visibility of error bars. Time was measured based on the number of frames between the introduction of the batch and reaching the melting temperature. The frame rate of the used camera is 125 Hz, so the accuracy of the measurement is 0.008 s, making it invisible in the figure.
  3. As suggested, the discussion of single measurements has been removed.
  4. A rectangular box depicting the standard deviation has been added to Figures 9 and 12. It will increase the informativeness of the figures.
  5. Distance units have been standardized throughout the article.

At the same time, we would like to inform you that the new version of this article has been submitted for professional language proofreading and has been certified.

Reviewer 4 Report

Dear Authors,

After I read the manuscript carefully, the document still many lacks data and experiments that need development and cannot be published in this form.

Many acronyms and symbols are used in the manuscript. Therefore, a list of nomenclature must be included.

ABSTRACT

Abstract is too small. The authors do not present many details about the proposed study. In addition to explaining the purpose of the study, it is necessary to explain in detail what electromagnetic levitation melting is, and to present the findings of this research. Therefore, the abstract needs to be rewritten.

INTRODUCTION

In the sentence “Nowadays, metals and alloys with high strength and high melting temperatures are increasingly used [1–3] in many applications.” What applications? This information needs to be detailed.

In this section, the authors could present the use of electromagnetic levitation melting in more detail. In this sense, the introduction should be improved with the inclusion of more information both on the traditional way of obtaining Ti alloys and on how electromagnetic levitation melting can become one of the forms of melting to be used.

2. MATERIALS AND METHODS

In this section, Figure 3 is unclear. How is it possible to know that the time was 89 seconds just by looking at the figure?

3. RESULTS AND DISCUSSION

The execution of the experiment and the theoretical basis do not allow to observe the progress proposed by the authors.

I recommend that a more careful analysis and discussion be carried out so that the manuscript presents the format to be considered for publication.

I suggest that the data presented to validate the simulation be accompanied by real tests that demonstrate the viability of the process presented.

CONCLUSIONS AND FUTURE RESEARCH

The authors comment that with the proposed simulations it will be possible to increase the efficiency of melting. In this case, why not perform theoretical and practical studies and discuss the results on the basis of the results obtained using electromagnetic levitation melting? In this way it would be possible to verify a comparison with the simulations and with the results obtained in practice.

Extensive editing of English language required.

Author Response

We thank you very much for the time you devoted to our article. We can see from your comments that you have thoroughly read our work and we appreciate it. Regarding your comments:

  • A list of acronyms used has been added at the very end of the article. This should help navigate between chapters.
  • The abstract and introduction chapters were rewritten according to the posted suggestions.
  • The description of Figure 3 has been modified to explain on what basis we know the time at which this frame from the video was taken. A frame number is 12282 one frame took 0.008 s, so the screen is 98.256 seconds after the batch was entered.
  • As you wrote the actual data should be put together with the simulation results. These can be found in Figures 9 and 12, in addition, information on the standard deviation of the measurements taken has been included.
  • Your ideas about experiments that could be further conducted on the basis of our work sound very interesting and we will consider them when taking further research directions.

At the same time, we would like to inform you that the new version of this article has been submitted for professional language proofreading and has been certified.

Reviewer 5 Report

The paper describes the analysis and measurement of magnetic induction heating-based equipment. The paper presents the details of the applied toolchain, the modelled geometry and the measurement.
This can be an interesting paper which deals with an important topic. However, the paper should be improved on the following points:

- abstract, this should be more specific, at least at the end, to present more concrete results from the paper.

 - There are many openly accessible benchmark problems in the literature, such as the TEAM 35 and TEAM 36 benchmark problems (10.1109/TMAG.2019.2951946), which has a good theoretical and measurement basis to compare the results or considering different methodologies for the manufacturing uncertainties: https://doi.org/10.1016/j.cam.2022.115021 . These are widely known problems, they should be referred in the first, motivational part of the paper, as similar research papers to show the state of the art of the problem.

 - Somehow it shuold be presented why the proposed benchmark is why scientifically interesting, or how can be used by other researchers. The novelty is missing from the introduction.

 - The proposed benchmark problem can be very interesting for other researchers to test their novel algorithms and compare their results precision or the performance of their tools. Is it possible to publish the model files and the measurement results in an openly accessible way?

- some words, like difficult a bit overused and its not only a grammatical problem, because it would be important to explain the difficulty more deeply

 - Please check the minor misstakes, like: bellow -> below in line 66

 There are some unnecessarily hard sentences, which can be clearified to help the reader to more easy to follow the text:

- The source of these problems is not only high hardness and negligible thermal conductivity, but also high melting point and high reactivity at high temperatures [7,8 ].

- The source of these problems is high hardness, negligible thermal conductivity, and high melting point and high reactivity at high temperatures [7,8 ]

 ----
 - The main issue with this type of melting is its general low energy efficiency; therefore, improvement and optimisation of this process is needed.

 - The main issue with this type of melting is its general low energy efficiency; therefore, this process needs improvement and optimisation.

Author Response

We thank you very much for the time you devoted to our article. We can see from your comments that you have thoroughly read our work and we appreciate it. Regarding your comments:

  • The abstract has been rewritten by us. Is more detailed, and includes information on the results achieved.
  • The benchmarks presented sound very interesting and could be a valuable addition to our introduction. We decided to include them in our work.
  • The novelty of the work is described in section 1.2
  • Of course, we would very much like our work to be useful to other researchers. Therefore, we include measurements of the process, and the geometry and parameters of the simulation model are described in the paper. This information should enable other researchers to reproduce and validate the model. The model itself has not been built based on standard software, but also on the author's codes which, in their current state, are not adapted for use by other researchers.
  • Thank you for your comments on hard-to-understand passages, they have been corrected.

At the same time, we would like to inform you that the new version of this article has been submitted for professional language proofreading and has been certified.

Round 2

Reviewer 1 Report

Thanks to the author's careful reversion, the revised manuscript can be accepted now.

Author Response

Thank you for your help in improving the article.

Reviewer 2 Report

The author has made some modifications, but the grid independence analysis is still not provided.

Author Response

Thank you for your vigilance, we must have overlooked this. For a description of grid independence analysis, see Section 3.3.

Reviewer 3 Report

The revised version addressed most of the comments, so I recommend publication in its present form.

Author Response

(The authors gave the same response as above.)

Reviewer 4 Report

Dear Authors,

I have reviewed the new version of the manuscript after the requested corrections. I have noticed that the authors took the time to improve the conducted study. I would like to say that I thank the authors for taking into account the presented corrections. The manuscript is now much better and can be considered for publication.

With best regards,

Reviewer

Author Response

(The authors gave the same response as above.)
